# Development of a Real-Time Pixel Array-Type Detector for Ultrahigh Dose-Rate Beams

**DOI:** 10.3390/s23104596

**Published:** 2023-05-09

**Authors:** Young Jae Jang, Tae Keun Yang, Jeong Hwan Kim, Hong Suk Jang, Jong Hwi Jeong, Kum Bae Kim, Geun-Beom Kim, Seong Hee Park, Sang Hyoun Choi

**Affiliations:** 1Research Team of Radiological Physics & Engineering, Korea Institute of Radiological & Medical Sciences, Seoul 01812, Republic of Korea; 2Department of Accelerator Science, Korea University, Sejong 30015, Republic of Korea; 3Center for ProtonTherapy, National Cancer Center, Goyang 10408, Republic of Korea

**Keywords:** 2D pixel array detector, real-time monitoring, collection efficiency, ultrahigh dose-rate beam, FLASH

## Abstract

Although research into ultrahigh dose-rate (UHDR) radiation therapy is ongoing, there is a significant lack of experimental measurements for two-dimensional (2D) dose-rate distributions. Additionally, conventional pixel-type detectors result in significant beam loss. In this study, we developed a pixel array-type detector with adjustable gaps and a data acquisition system to evaluate its effectiveness in measuring UHDR proton beams in real time. We measured a UHDR beam at the Korea Institute of Radiological and Medical Sciences using an MC-50 cyclotron, which produced a 45-MeV energy beam with a current range of 10–70 nA, to confirm the UHDR beam conditions. To minimize beam loss during measurement, we adjusted the gap and high voltage on the detector and determined the collection efficiency of the developed detector through Monte Carlo simulation and experimental measurements of the 2D dose-rate distribution. We also verified the accuracy of the real-time position measurement using the developed detector with a 226.29-MeV PBS beam at the National Cancer Center of the Republic of Korea. Our results indicate that, for a current of 70 nA with an energy beam of 45 MeV generated using the MC-50 cyclotron, the dose rate exceeded 300 Gy/s at the center of the beam, indicating UHDR conditions. Simulation and experimental measurements show that fixing the gap at 2 mm and the high voltage at 1000 V resulted in a less than 1% loss of collection efficiency when measuring UHDR beams. Furthermore, we achieved real-time measurements of the beam position with an accuracy of within 2% at five reference points. In conclusion, our study developed a beam monitoring system that can measure UHDR proton beams and confirmed the accuracy of the beam position and profile through real-time data transmission.

## 1. Introduction

In recent years, active research has been conducted on radiotherapy (RT) that uses ultrahigh dose rates (UHDRs). The UHDR beam maximizes treatment efficiency and minimizes the damage to healthy tissues that occurs with high doses at rates ≥40 Gy/s [1,2,3,4,5,6,7]. Furthermore, the use of high-intensity radiation effectively destroys tumor cells, enabling improved therapeutic effects compared to those of conventional RT methods. In addition, the treatment duration is short because a valid therapeutic effect is achieved from only a few treatment sessions [8,9,10]. However, because UHDR beams are characterized by single, precise high-dose irradiation within a short time frame, there is a need for a detector that is capable of rapid and accurate beam profiling before a beam can be used for treatment.

Detectors commonly used in RT are ionization chambers, which are mainly used to measure the dose of the radiation beam, and the Matrixx, or diode-type, detector, which provides 2D measurements for beam profiling [11,12]. However, conventional, commercially available detectors used for the measurement of UHDRs can result in high beam loss owing to a decrease in the collection efficiency caused by the ion recombination effect. Therefore, they are unsuitable for measuring beams with high dose rates [13].

For 2D detection, strip-type detectors are widely used because of their high detection efficiency caused by their long strip length, which enables the detection of several particles and narrow strip intervals for determining precise particle positions [14]. The recent introduction of the proton and electron beam FLASH-RT has prompted the development of 2D detectors that are capable of measuring beams with high dose rates. Yunjie et al. developed a strip-type detector to measure a 250 MeV proton PBS beam at a high dose and resolution [15]. Zou et al. measured a current of up to 350 nA in a 226.29 MeV FLASH proton beam [16]. However, the strip-type detector used in both studies prevented the accurate measurement of beams that were simultaneously received by the irradiated surface. Furthermore, data acquisition at high speeds was limited.

Accurate beam profiling is possible using a 2D-pixel-type detector, owing to its high rate of particle detection via the signal received by each pixel as well as its high positional accuracy. Therefore, pixel detectors are widely used in quality assurance (QA) measurements for patients [17]. However, the high cost of production for large-area measurements and the problem of ion recombination caused by the presence of ion pairs in the detector when measuring UHDR beams make achieving accurate measurements challenging [18]. Therefore, for the accurate, real-time beam profiling of UHDR beams, using a high-speed data processing capacity for beams irradiated in short-time spans and taking accurate measurements of the dose by minimizing the ion recombination effect that arises in pixel-type detectors are required [19].

This study developed a pixel array-type detector with data acquisition (DAQ) for high-speed data processing to achieve accurate measurements of UHDR beams. The ion recombination effect, a limitation of conventional detectors, was minimized by developing the collection efficiency, which was calculated by adjusting the high voltage and gap-frame thickness in the detector; this also minimized beam loss. Finally, the novel detector was validated by verifying whether real-time data processing could be achieved on the UHDR beams that were being irradiated to the pixel detector through beam profiling.

## 2. Materials and Methods

### 2.1. Development of the Beam Monitoring System

A beam monitoring system is a device for measuring the intensity and position of the cross-section of an irradiated ion beam. The system used in this study comprised a 2D-pixel detector and DAQ component. The 2D-pixel detector was layered from left to right with a window, pixel-type electrodes, a gap, high-voltage electrodes, and another window (Figure 1). There were 256 pixel-type electrodes in the system with a dimension of 4 mm × 4 mm; the pixel interval was 0.125 mm. The maximum level of measurement was 65 mm × 65 mm. The detector was designed to allow the space between pixel electrodes and high-voltage electrodes to be adjusted by 2–5 mm using an independently fabricated gap. The gap frame was produced using a 3D printer with PLA material, which is an insulator and has a melting point higher than 200 degrees. To adjust the gap, a gap frame with the desired thickness was manually inserted between the cathode electrode and high-voltage electrode. The maximum voltage bias was 2000 V (Table 1).

The DAQ process comprised software programmed with Labview [20] to enable the real-time monitoring of the beam and DAQ board, which received signals. The DAQ board was connected to the detector, which measured the beam current, position, and profile, to perform data collection and transmission. The current from the detector upon beam irradiation was measured using the charge integration method with an AFE0064 chip (Texas Instrument, Dallas, TX, USA) [21]. The measured values were converted using an analog-to-digital converter (ADC) and stored at a 40 μs frequency. Subsequently, the values were transmitted in real time to a PC connected via TCP/IP (Figure 2). The analog interface of the DAQ board was able to accept input signals in the range of 0 to +24 V and generate high voltages in the range of 0 to 2000 V based on the input value. The digital interface controlled the operation of the DAQ board during start and stop times. The FPGA was reprogrammable, allowing the current measurement settings to be customized depending on the type of beam measured. The measured beam data were stored in real time using 4 GB DDR3 DRAM. Using a PC, intensity values from each of the 256 pixels were visualized in real time using an independent software programmed with Labview. The beam position and dispersion was also checked in real time. For data extraction, the maximum recordable time per file was 1000 s and the gain value at measurement could be controlled at 0–7 to eliminate noise and collect amplified values for the target signals (Table 2).

### 2.2. Monte Carlo Simulation

Boag’s method was used to calculate the collection efficiency [22], and the analysis was based on the ion-pair values generated in 1 s. The Particle and Heavy Ion Transport System (PHITS) Monte Carlo code [23] was used to estimate the charge deposited per pixel, energy transferred to the detector, and saturation charge. The T-deposit tally was used to calculate the number of ion pairs generated in each electrode and for each gap thickness, as well as the deposited charge value. In addition, the T-track tally was used to confirm proton beam tracking in the pixel detector. The beam irradiation surface comprised 256 pixels on the xy plane, each pixel being 4 mm × 4 mm × 35 μm, which are the same dimensions as those of the fabricated pixel array detector. To generate adequate ion-pair values, the gap between the pixel-type electrodes and high-voltage electrodes was adjusted to 2, 4, and 5 mm. To compare the beam energy during simulation with the actual measured values, currents of 10, 40, and 70 nA were applied to a 45 MeV proton beam, in addition to a 1.842 nA and 226.29 MeV proton beam.

The collection efficiency was calculated as follows:(1)$$f_{g}=\frac{Q\left(V\right)}{Q_{sat,g}}=\frac{1}{1+\lambda_{g}Q_{sat,g}/V^{2}},\;\lambda_{g}=\frac{\alpha d^{4}}{6ek_{1}k_{2}\nu T}$$
where *Q_sat,g_* is the saturation value of the charge for general recombination, which should be obtained; e is the electron charge, which is a constant; *k*_1_ and *k*_2_ are the mobilities of the positive and negative ions, respectively; α is the recombination constant; and *T* is the pulse duration, which varies according to experimental conditions. The volume of the fabricated detector is reflected by *v*, which is the volume of the detector. Considering that the collection efficiency is mostly influenced by *V*^2^; the square of the voltage; and *d*^4^, the 4th power of the electrode gap that indicates the thickness of the gap or space between electrodes, a higher level of collection efficiency can be achieved by decreasing the gap and increasing the voltage.

### 2.3. Experiment Setup

#### 2.3.1. Collection Efficiency

To minimize beam loss from the ion recombination effect prior to beam profiling, an experiment was conducted to estimate the collection efficiency of a beam based on its voltage and gap adjustment. Two beam conditions were applied in this experiment. First, 10, 40, and 70 nA currents were used for a 45 MeV energy beam generated using the MC-50 cyclotron (Korea Institute of Radiological and Medical Sciences (KIRAMS, Seoul, Republic of Korea)). The beam energy of the MC-50 cyclotron was 18.0 to 50.5 MeV, in which a current of 1 nA to 60 µA was available. A collimator with a 10 mm diameter was attached to the beam output to control the beam size at 1 σ = 4.25 mm. Second, a 1.842 nA current was used for a 226.29 MeV energy PBS proton beam at 1 σ = 5.0 mm produced using the proton therapy machine (National Cancer Center (NCC, Goyang, Republic of Korea)). The distance between the beam source and detector was 100 cm in both conditions. While the detector gap was fixed at 2, 4, and 5 mm, the high-voltage bias was adjusted to be in the range of 0–1500 V to determine the threshold voltage and gap conditions that minimize beam loss. Prior to the experiment, the detector was set in a place that situated the beam irradiation at its center (Figure 3).

#### 2.3.2. Checking Beam Size and Position

To check the device’s capacity for measuring FLASH proton beams, a 45-MeV energy beam was irradiated to the pixel detector at 10, 40, and 70 nA currents from the MC-50 cyclotron. To check the dose, a Gafchromic HD-V2 film [24], which allows up to 1000 Gy, was placed at the same position as the beam irradiation. The irradiated film was analyzed using the DoselabPro software to measure the beam profile [25]. The dose values obtained from the film and the intensity values measured using the pixel detector were then matched, and the measured intensity values were converted into a dose. For the 1.842 nA current applied to a 226.29-MeV energy beam generated using the NCC proton therapy machine, the beam’s irradiation was based on five reference points: (−15,19), (19,15), (0,0), (−19,−15), and (15,−19) (mm). The beam’s position was determined at the pixel detector, and a comparative analysis was performed by irradiating the beam to the Matrixx (Matrixx PT, IBA Dosimetry, Herndon, VA, USA) [26] at the five points to evaluate the accuracy of the position at the pixel detector. The Matrixx has a total measurable area of 24 cm × 24 cm and a resolution of 7.6 mm and is capable of measuring doses up to 40 Gy/min.

## 3. Results

### 3.1. Beam Condition

The results obtained from the PHITS simulation are presented in Figure 4. Figure 4a shows the irradiation of the proton beam at the pixel detector. The plots of the energy transferred to each pixel and ion pair generated at the gap were obtained to calculate the saturation charge at the detector, as shown in Figure 4b,c. To obtain the saturation charge value, the number of ion pairs obtained in each pixel from the simulation results was multiplied by the current value used in the experiment, and the collection efficiency was obtained by entering the d and V values into Equation (1).

Figure 5 shows the collection efficiency values according to d and V, where the empty shape represents the results obtained from the PHITS simulation, and the filled shape represents the saturation charge value measured during the experiments. The saturation charge, calculated using Equation (1), was 3.34 × 10^−7^ C/s for a 2 mm gap, 10 nA current, and 45 MeV energy proton beam, which increased to 8.36 × 10^−7^ C/s as the gap increased to 5 mm, which was obtained from the PHITS simulation (Figure 5a). The obtained value from the PHITS simulation was 0.992 for a gap of 5 mm and voltage of 1500 V, which indicates a ~0.4% beam loss. The collection efficiency approached 1 as the voltage increased within the range of 0–1500 V in both the simulation and measurements. As the current increased to 40 and 70 nA for a 2 mm gap and 45 MeV energy beam, the saturation charge obtained from the PHITS simulation increased to 1.14 × 10^−6^ C and 1.87 × 10^−6^ C, respectively, and a 2–5% beam loss was observed as the gap increased to 4 and 5 mm, unless the voltage increased to ≥1000 V (Figure 5b,c). Figure 5d shows the values obtained from the experiment for a 226.29 MeV proton beam, used to determine the conditions that would not cause a loss in collection efficiency before the evaluation of positional accuracy. Because the current for the NCC proton beam was low, the beam loss was ≤0.5% at 500 V for 2, 4, and 5 mm gaps. For a voltage of 1000 V and gap of 2 mm, the beam loss was below 0.1% for all the simulation and experimental values.

### 3.2. Beam Profiles

The signals received by the DAQ were the intensity values. The film calibration curve was drawn to convert these measured beam values to dose values, which were used to determine the proton beam dose across varying conditions (Figure 6a). The results indicate that the UHDR beam output for a 45-MeV energy proton beam was 76–77, 142–149, 209–233, 260–303, and 309–375 Gy/s with applied currents of 10, 25, 40, 55, and 70 nA, respectively (Figure 6b).

The beam profiles for 10, 40, and 70 nA currents applied to a 45 MeV energy proton beam generated using the MC-50 cyclotron (i.e., the FLASH proton beam conditions) were analyzed. Figure 7 shows the results for a 10 nA current. Figure 7a shows the beam profile obtained from the software in real time, whereas Figure 7b shows the plot of the dose values after they were converted from the film irradiation at the same position as the pixel detector. Figure 7c,d show the beam profiles for the X and Y positions, respectively. In comparing the distributions for Position X, the detector values were found to be Xc = −1.45 mm and 1σ = 7.555 mm for the variations of 0.4 and 0.16 mm, respectively, and their respective film values were found to be Xc = −1.85 mm and 1σ = 7.715 mm (Figure 7c). For Position Y, the detector values were found to be Yc = −1.04 mm and 1σ = 7.736 mm for the variations of 0.39 and 0.15 mm, respectively, and their respective film values were found to be Yc = −1.43 mm and 1σ = 7.889 mm (Figure 7d). For both Position X and Y, the peak of the Labview value was high and σ was low, whereas the deviation was smaller for Y than that of X.

To evaluate the positional accuracy, a 226.29 MeV and 1.842 nA energy beam generated from the NCC proton therapy machine was used, and the beam distribution was obtained after collimator. A gap of 2 mm and a voltage of 1000 V were used to minimize beam loss for the collection efficiency. Figure 8a shows the beam profile from the pixel detector upon beam irradiation as monitored in real time using Labview. Figure 8b shows the plot based on the data stored by the DAQ. Figure 8c,d show the measurements taken using the Matrixx and the respective comparative analysis with the positional data from the pixel detector.

Table 3 presents the calculated values of the position at five reference points of beam irradiation based on the intensity values received by each of the 256 pixels at the pixel detector as well as the measured values from the Matrixx. The reference point with the largest deviation from the measured value at the pixel detector was (19,15) with σ_x_ = 0.5 mm and σ_y_ = 0.7 mm. For the center point (0,0), the deviation was ≤0.1 mm. The error range for the measurements taken using the Matrixx at the five reference points was 0.1–1.6 mm, indicating a larger deviation than that of the pixel detector.

## 4. Discussion

The results obtained from the five reference point measurements using the Matrixx and pixel detector were found. However, to identify the cause of errors for the five reference points used for evaluating the positional accuracy, a binary file stored in the DAQ was extracted and analyzed. Among the five reference points, the one with the largest margin of error was (15,−19) with 14.4833 mm on the x coordinate, which deviated from the 15.00 mm of the conventional device by 0.5167 mm. In contrast, the reference point with the smallest margin of error was (0,0) deviated by 0.0283 mm on the x coordinate. Through analyzing each point for the beam position, it was seen that the standard deviation on the x and y axes was in the range 0.4055–0.4989 and 0.4232–0.4946, respectively, with a higher precision for the x values than for the y values. The distance between each reference point was estimated, and the relative interval across the coordinates was analyzed. The measured interval was 33.65 mm, which was ~0.58 mm shorter than the reference value (34.23 mm), indicating that the distance from the pixel detector to the proton beam was shorter than the set value and varied across the four gap areas. Therefore, the presence of rotation and tilt upon the detector alignment was conjectured.

A comparative analysis of the values from the two devices to determine whether the beam size was small upon irradiation could not be performed because the beams overlapped in the analysis of the five reference beams, owing to the 7 mm pixel size of the Matrixx. An advantage of the proposed pixel detector is that the real-time information of each pixel and the stored data can be used to determine the profile and position of the beam.

To ensure accurate measurements of FLASH beams, UHDR beam distributions and doses should be simultaneously measured. The strip-type detector developed by Zou et al. minimized the ion recombination effect by setting the gap at 1 mm to take measurements of the FLASH beam [16]. In contrast to previous studies, the gap frame thickness could be adjusted in our study, so that when the dose rate of the target beam was very high, the gap frame thickness could be decreased to reduce the ion recombination effect. Among the measured FLASH beams, the proton beam with a 45-MeV energy, 10-nA current, and 70-G/s dose rate reduced the ion recombination effect when the gap was 5 mm and the voltage was 1000 V. Consequently, the proposed system can accurately determine inherent beam data because the background noise from large signals can be removed as a result of the higher levels of ion-pair formation for a 5 mm gap compared to that for a 1 mm gap.

Furthermore, as the detector was developed to be a pixel type, in contrast to the conventional strip-type 2D detectors, this novel detector is suitable for QA measurements for each patient as well as beam profiling before applying FLASH-RT in patient treatment. For beam profiling, the proposed system can perform rapid data processing because the intensity values are received in real time for irradiated beams with a 40 μs frequency, and measurements can be taken for proton, carbon, and electron beams.

## 5. Conclusions

This study proposed a pixel array-type detector with a monitoring system for conducting accurate real-time measurements of UHDR proton beams. The utility of the novel detector was tested through a comparative analysis using the PHITS Monte Carlo code and proton beam measurements. The dose rate of a proton beam with a 45 MeV energy and 70 nA current was found to be ≥300 Gy/s at the center of the beam, and the pixel detector enabled real-time monitoring. The accuracy of the position determined by the detector was verified by analyzing the coordinates of five reference points.

The conditions for a <1% ion recombination effect in the detector were identified by adjusting the gap and voltage. For a 45-MeV FLASH proton beam with a 10 nA current and 70 Gy/s dose rate at the center of the beam, beam loss was <1% for gaps of 2, 4, and 5 mm at a voltage of 1000 V.

The proposed 2D-pixel detector has 256 pixels of 4 mm × 4 mm. To improve the resolution, a high-resolution detector with a large area will be developed. Furthermore, for measurements in FLASH-RT with high-surge current electron beams, the acceptable charge for the DAQ will be set at a high level.

## Figures and Tables

**Figure 1 sensors-23-04596-f001:**
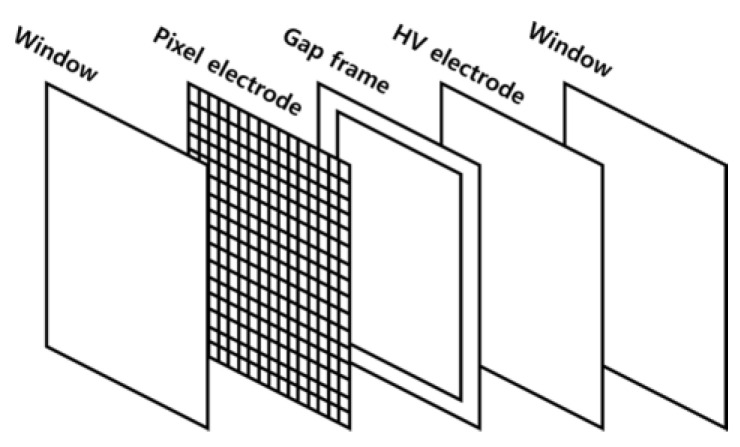
Schematic of the pixel detector.

**Figure 2 sensors-23-04596-f002:**
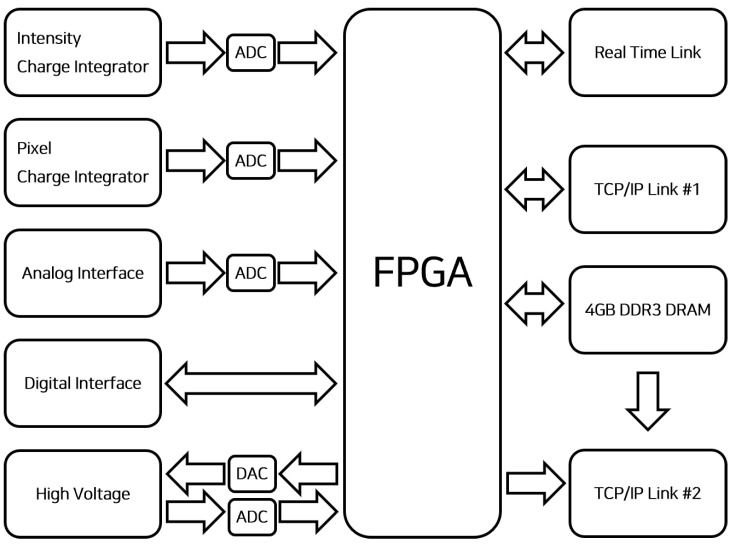
Flow chart of the DAQ board.

**Figure 3 sensors-23-04596-f003:**
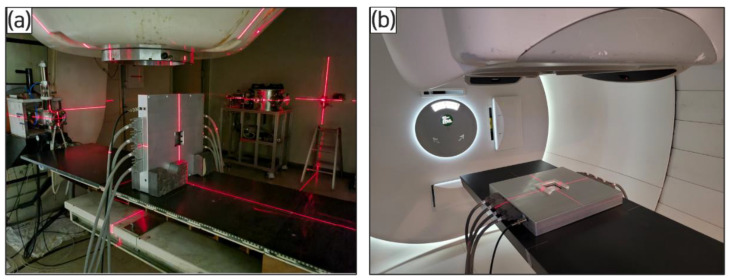
Photos of beam alignment at center of detector before experiment; (**a**) MC-50 cyclotron of Korea Institute of Radiological and Medical Sciences and (**b**) proton therapy machine of National Cancer Center. The distance between the beam source and detector was 100 cm.

**Figure 4 sensors-23-04596-f004:**
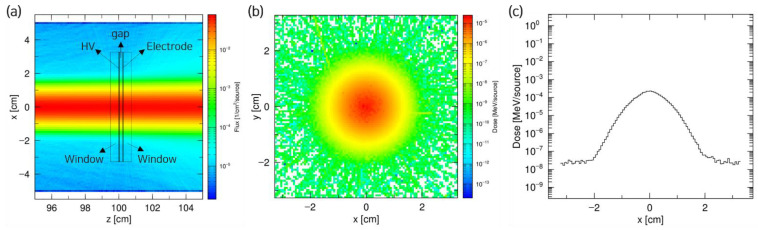
PHITS simulation results for the 45 MeV proton beam irradiation at the pixel detector. (**a**) Beam tracking in pixel detector; (**b**) beam shape in the gap; and (**c**) dose in the gap area.

**Figure 5 sensors-23-04596-f005:**
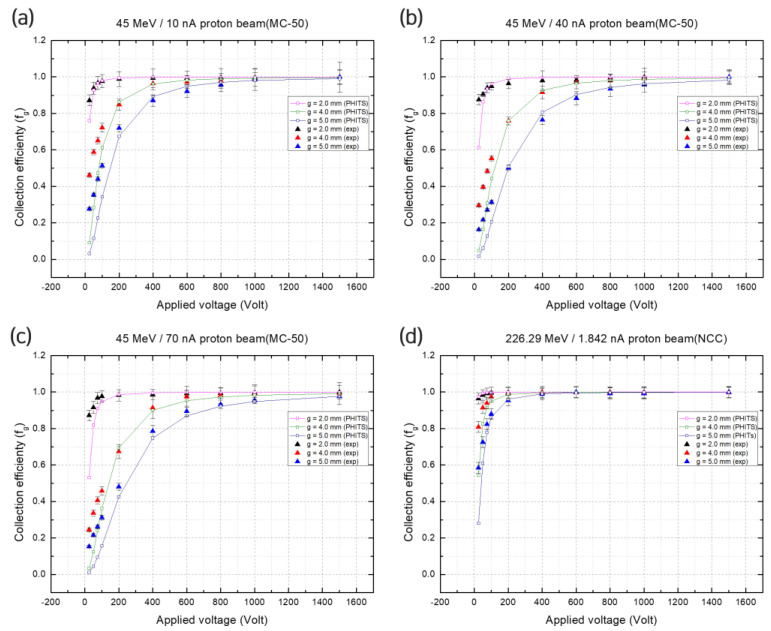
Collection efficiency of pixel detector. (**a**) 45 MeV/10 nA MC-50; (**b**) 45 MeV/40 nA MC-50; (**c**) 45 MeV 70 nA MC-50; and (**d**) 226.29 MeV/1.842 nA NCC. In each image, the empty and filled shapes are the PHITS simulation and experimental result values, respectively.

**Figure 6 sensors-23-04596-f006:**
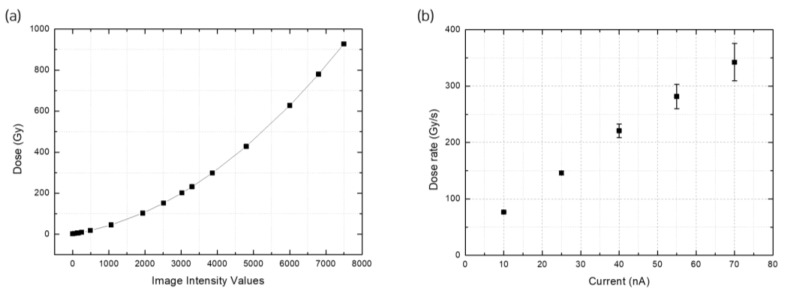
Dose-rate conversion. (**a**) Calibration curve and (**b**) dose rate for a 45-MeV energy proton beam.

**Figure 7 sensors-23-04596-f007:**
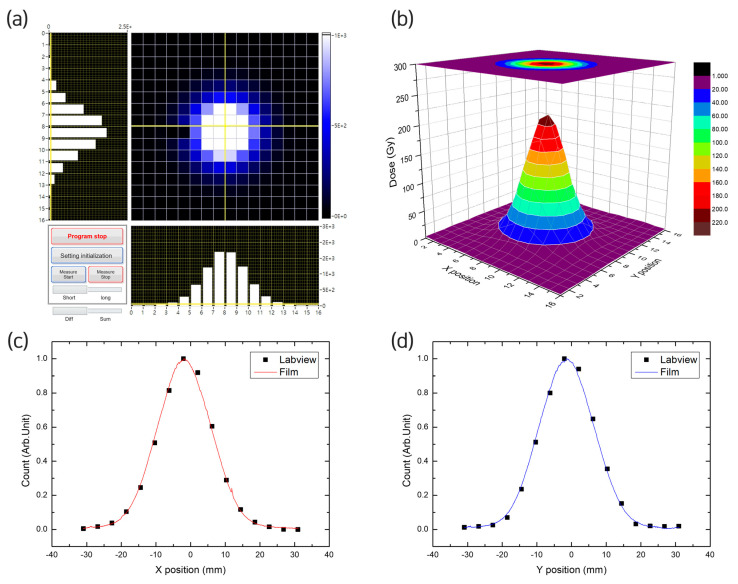
MC-50 cyclotron beam profiling. (**a**) Real-time detection of the beam shape on the Labview software; (**b**) visualization of the dose calculated based on film irradiation; (**c**) Labview and film beam profiling at Position X; and (**d**) Labview and film beam profiling at Position Y.

**Figure 8 sensors-23-04596-f008:**
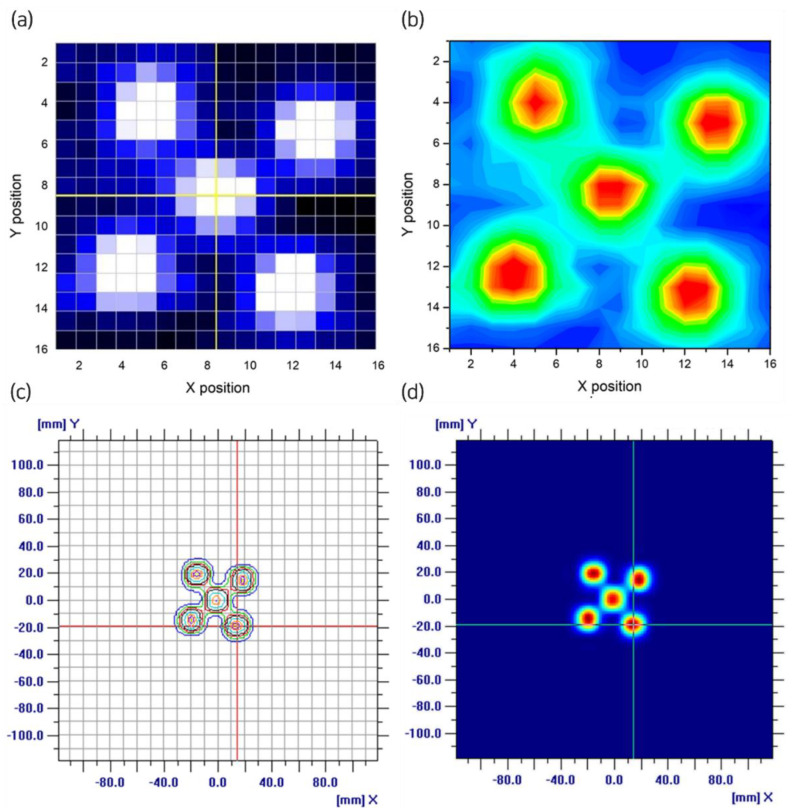
Measurements at five reference points using pixel detector and Matrixx. (**a**) Cumulative values checked in real time at pixel detector; (**b**) plot based on data stored by DAQ; (**c**) Matrixx values at five reference points; and (**d**) distribution of positional accuracy checked using Matrixx.

**Table 1 sensors-23-04596-t001:** Composition and specification of the pixel detector.

Parts	Material Information	Specification
Mylar window	Mylar film	0.5 μm thickness
Electrode (cathode)	1.6 mm PCB layer +17.5 μm copper (both side)	(4.0 × 4.0) mm^2^ 256 pixels 125 μm between pixels
Gap (adjustable)	3D printed PLA	2.0–5.0 mm gap
Electrode (high voltage)	1.6 mm PCB layer +17.5 μm copper	Maximum High voltage: 2000 V
Mylar window	Mylar film	0.5 μm thickness

**Table 2 sensors-23-04596-t002:** Composition and specification of the DAQ board.

Part	Specification
Intensity current	Output: 40 μs
Pixel current	Output: 160 μs
Maximum data	1000 s (8,388,608 frames)
High voltage	0–2000 V
Gain	0–7
Current range	1–447 nA (20 μs)
Maximum charge	9.5 pC
Chip	AFE0064

**Table 3 sensors-23-04596-t003:** Evaluation of the positional accuracy based on five reference points.

Reference Point	Matrixx				Pixel Detector			
(x,y) (mm)	P_x_	P_y_	σ_x_	σ_y_	P_x_	P_y_	σ_x_	σ_y_
(−15,19)	−15.4	18.8	0.4	0.2	−14.9	18.6	0.1	0.4
(19,15)	18.5	13.9	0.5	1.1	18.5	14.3	0.5	0.7
(0,0)	−1.0	−0.4	1.0	0.4	0.0	−0.1	0.0	0.1
(−19,−15)	−19.4	−14.8	0.4	0.2	−18.9	−14.8	0.1	0.2
(15,−19)	13.4	19.1	1.6	0.1	14.5	−19.1	0.5	0.1

## Data Availability

Not applicable.

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
