# Peer review of "Development of a Real-Time Pixel Array-Type Detector for Ultrahigh Dose-Rate Beams"

_sensors, 2023, doi:10.3390/s23104596_

Round 1

Reviewer 1 Report

In this paper, a pixel array detector with adjustable gaps and a data acquisition (DAQ) system are developed to evaluate its effectiveness in real-time measurement of UHDR proton beams. The UHDR beam was measured using a MC-50 cyclotron at 70nA current at 45MeV energy. The collection efficiency of the detector and the real-time measurement accuracy of the beam position were determined by experimental measurements and simulations. The results show that the beam monitoring system can accurately measure the UHDR proton beam and transmit the beam position and profile data in real time. This work is very interesting. Therefore, this manuscript is worth publishing in sensors. However, the author needs to address my concerns as follows.

1.       What kind of proton beam does the MC-50 cyclotron used in this study produce? How does this study determine the collection efficiency of the detector and the real-time measurement accuracy of the beam position?

2.       The English expression may be modified appropriately, and the references should be checked carefully.

3.       How to solve the beam loss caused by traditional pixel detector? What kind of detector and data acquisition system has been developed in this study?

4.       The results of the experiment are not clearly expressed and need to be modified.

5.       What is UHDR radiotherapy? Why do we need to measure the UHDR proton beam?

Reviewer 2 Report

The authors have presented a highly intriguing study that describes a detector equipped with a monitoring system for accurate, real-time dose measurements. 

The efficacy of this innovative detector was evaluated using both simulations and experiments. Overall, this manuscript represents a significant step forward in the development of the FLASH-RT technology, allowing for improved characterization and leveraging of this technology.

While the manuscript is well-written and has a precise aim, further refinement is required to make it publishable, particularly clarifying the methodology used to obtain the results. 

Additionally, the literature review is lacking in some areas that may be of great interest to readers interested in this topic.

One general comment regarding the manuscript is that the citation style is unconventional. It is recommended that citations be placed at the end of sentences before the period. Furthermore, the use of "author name and et al." should be avoided.

1.     Introduction:

-       Several interesting paper/review papers that would be interesting for the topic treated have been omitted in state of the art. 

I would suggest taking a look at them:

o   Romano, Francesco, et al. "Challenges in dosimetry of particle beams with ultra-high pulse dose rates." Journal of Physics: Conference Series. Vol. 1662. No. 1. IOP Publishing, 2020.

o   Lourenço, Ana, et al. "Absolute dosimetry for FLASH proton pencil beam scanning radiotherapy." Scientific Reports 13.1 (2023): 2054.

o   Poppinga, Daniela, et al. "VHEE beam dosimetry at CERN Linear Electron Accelerator for Research under ultra-high dose rate conditions." Biomedical Physics & Engineering Express 7.1 (2020): 015012.

o   Chaudhary, Pankaj, et al. "Radiobiology experiments with ultra-high dose rate laser-driven protons: methodology and state-of-the-art." Frontiers in Physics 9 (2021): 624963.

2. Materials and Methods

- The referencing style of figures and tables is unusual everywhere.

- Table 1 is hard to read

- More justification is needed on why certain materials are needed/used. An example of interest is using the highly unreliable PLA for work related to UHDR.

- Reference and citation from the chip used, and its previous application would be beneficial for the paper

- The paper would largely benefit from more information about the actual DAQ configuration, including the FPGA.

- In line 133, it needs to be clarified the gap used.

- Missing information about the mechanical system detail that might be relevant 

- In the chapter Montecarlo simulation, detailed information are missing.

- The situation described from 176-179 is not ideal at all. Why were the films not calibrated using an independent and calibrated source?

- Also, were the films installed at the same time as the irradiation or at a different time? In case not, what is the reproducibility expected?

- Gafchromic HD-V2, DoseLabPro,  and Matrixx deserve a proper reference for readability.

3. Result

-           From 186 to 188 style of referencing Fig. 4 is different. Furthermore, it is not clear the meaning of Fig. 4c.

-           The maximum voltage value is referred to be 1.5kV. Instead was previously mentioned as a 2kV. Please clarify.

-           All over subsection 3.1 the writing style needs to be fixed. As well as is unclear which results come from simulation and which from the experiment.

-           Also, in the picture, it would be beneficial to have the measurement point in one style and the simulation in another style.

-           From 249 to 254, why dispersion might have been swapped with the position. Please clarify them.

-           Could u please clarify the beam distribution in Figure 8? Was this obtained after a collimator?

-           What are the distances used in the experiments? A schematic picture would be beneficial.

5. Discussion

-           From 288 to 289, you refer to commercially available devices. Do you mean films? 

-           Also, it looks like you have missed one section. The Result is section number 3, and the Discussion is section 5.
